# A Modified Complex Variational Mode Decomposition Method for Analyzing Nonstationary Signals with the Low-Frequency Trend

**DOI:** 10.3390/s22051801

**Published:** 2022-02-24

**Authors:** Qiuyan Miao, Qingxin Shu, Bin Wu, Xinglin Sun, Kaichen Song

**Affiliations:** College of Biomedical Engineering and Instrument Science, Zhejiang University, Hangzhou 310027, China; 11715040@zju.edu.cn (Q.M.); 11915040@zju.edu.cn (Q.S.); wubinzju@zju.edu.cn (B.W.); kcsong@zju.edu.cn (K.S.)

**Keywords:** variational mode decomposition, complex-valued signal processing, complex variational mode decomposition, nonstationary signal processing, low-frequency trend

## Abstract

Complex variational mode decomposition (CVMD) has been proposed to extend the original variational mode decomposition (VMD) algorithm to analyze complex-valued data. Conventionally, CVMD divides complex-valued data into positive and negative frequency components using bandpass filters, which leads to difficulties in decomposing signals with the low-frequency trend. Moreover, both decomposition number parameters of positive and negative frequency components are required as prior knowledge in CVMD, which is difficult to satisfy in practice. This paper proposes a modified complex variational mode decomposition (MCVMD) method. First, the complex-valued data are upsampled through zero padding in the frequency domain. Second, the negative frequency component of upsampled data are shifted to be positive. Properties of analytical signals are used to get the real-valued data for standard variational mode decomposition and the complex-valued decomposition results after frequency shifting back. Compared with the conventional method, the MCVMD method gives a better decomposition of the low-frequency signal and requires less prior knowledge about the decomposition number. The equivalent filter bank structure is illustrated to analyze the behavior of MCVMD, and the MCVMD bi-directional Hilbert spectrum is provided to give the time–frequency representation. The effectiveness of the proposed algorithm is verified by both synthetic and real-world complex-valued signals.

## 1. Introduction

Nonstationary signal processing methods have attracted considerable interest over the last few decades. Typical methods such as short-time Fourier transform (STFT) [1], the continuous wavelet transform (CWT) [2], S transform (ST) [3], and Wigner–Ville distribution (WVD) [4] can provide a time–frequency or time–scale representation for analyzing nonstationary signals. However, these algorithms are based on specific bases. Their performance depends on the selection of basis and is restricted by the uncertainty principle [5]. In order to break the limitation of the uncertainty principle, the empirical mode decomposition (EMD) [6] was proposed by Huang et al. as a data-driven signal decomposition method. EMD can recursively decompose a nonstationary signal into data-dependent basis functions named the intrinsic mode function (IMF) [7]. Consequently, the instantaneous frequency characteristics of the nonstationary data can be directly obtained from the IMFs [7]. However, EMD lacks mathematical supports and has limitations such as mode mixing. Therefore, variational mode decomposition (VMD) [8] was proposed in 2014 as a newly developed method alternative to EMD, fully established on a mathematical framework [9,10]. It can adaptively decompose a multi-component signal into several sub-signals and is not restricted by the uncertainty principle. VMD has been proven to outperform EMD and many other time–frequency analysis techniques in many applications, such as detection, separation and fault diagnosis [8,11,12,13,14]. However, both EMD and VMD are designed for real-valued data, which do not satisfy the meet of complex-valued signal processing. Since complex-valued data are widely used nowadays [15,16,17,18], it is vital to extend existing real-valued signal decomposition approaches to complex-valued signals.

Several time–frequency decomposition techniques have been studied to deal with the nonstationary multivariate signals [19,20,21]. Bivariate EMD [22] proposed in 2007, multivariate EMD proposed [23] in 2010, and multivariate VMD [24] proposed in 2019 can be regarded as extensions of the mode decomposition methods to complex-valued data. However, this class of complex extension methods pays more attention to analyzing different channels of the data. Since complex-valued data are different from multivariate data, these approaches ignore the physical meaning of complex-valued data and lose the mutual information between the real and imaginary parts of the complex-valued data [25]. Therefore, another extended EMD method for complex-valued data named complex EMD (CEMD) [26] was developed by Toshihisa Tanaka and Danilo P. Mandic. The relationship between the Fourier spectrum and the analytical signal was fully used in CEMD. Similar ideas have been applied for extending standard VMD to the complex domain [25,27]. The complex extension of VMD (CVMD) divides the complex-valued signal into positive and negative frequency components. According to the properties of the analytical signal [28], two real-valued signals are generated from the positive and negative frequency components, respectively. Standard VMD is then applied to the two real-valued signals, and a set of sub-signals can be produced. Hilbert transform is finally used to get the complex-valued form of the sub-signals. However, this kind of approach encounters difficulties when the data contain low-frequency trends. Specifically, the low-frequency component may be around zero frequency and distributed over both positive and negative frequencies. CVMD separates it into two parts directly. Therefore, prior knowledge is required to rearrange the separated signal for reconstructing the low-frequency mode. Even if the reconstruction is successful with correct rearrangement, there is energy loss, which will be analyzed in Section 3. Another drawback in CVMD is related to the decomposition number. The decomposition number is needed as a priori parameter in VMD. Since CVMD separates the data into two parts, and the VMD algorithm should be applied twice, each of the parts’ decomposition number is required prior. Existing methods assume the number of sub-signals in the negative and the positive frequency domain is identical, while it is not always applicable in practice.

Complex-valued nonstationary data with low-frequency components are common in many areas, and there are great interests in analyzing them. For example, the low-frequency component in complex-valued time series represents a trend over time that can reveal a specific pattern [25,26]; in array processing, the low-frequency component in complex-valued array data represents the broadside direction of the incoming wave [29]; in wind forecasting, studying the low-frequency components in complex-valued wind data can help to forecast the wind profile [30]. However, there are no available methods to analyze complex-valued nonstationary data with low-frequency trends. On the one hand, VMD, one of the best novel nonstationary signal analysis methods, as well as its modified versions, is not suitable for complex-valued data. On the other hand, the existing complex extension of the VMD method, known as CVMD, has drawbacks when analyzing low-frequency signals. Moreover, no modified approaches focus on this problem to the authors’ knowledge. Therefore, proposing a modified CVMD method for analyzing complex-valued data with low-frequency trends is necessary and significant.

This paper proposes a modified complex variational mode decomposition method (MCVMD), giving a new and simple way to extend VMD to the complex domain, suitable for analyzing complex-valued nonstationary data with low-frequency trends. The rest of this paper is organized as follows. Firstly, Section 2 briefly recalls the related work about VMD and CVMD. Secondly, the proposed algorithm and its Hilbert spectrum and filter bank property are described in Section 3. Then, Section 4 provides the simulation results and analysis to verify the performance of the proposed method. At last, the conclusion is done in Section 5.

## 2. Related Work

### 2.1. Variational Mode Decomposition

VMD was first proposed by Dragomiretskiy and Zosso in 2014, a novel signal decomposition method for time–frequency analysis [8]. It can non-recursively decompose a real-valued multi-component signal f into K number of quasi-orthogonal band-limited sub-signals $u_{k}$, also called band-limited intrinsic mode functions (BLIMFs). The VMD method is realized through the constrained variational problem:(1)$$\underset{\left\{u_{k}\right\},\left\{\omega_{k}\right\}}{\min}\{\sum_{k=1}^{K}\|\partial_{t}\left[\left(\delta \left(t\right)+\frac{j}{\pi t}\right)*u_{k}\left(t\right)\right]e^{-j\omega_{k}t}{\|}_{2}^{2}\},\;\mathrm{subject}\;\mathrm{to}\;\sum_{k=1}^{K}u_{k}=f,$$
where $\{u_{k}\left(t\right)\}=\left\{u_{1},\ldots ,u_{K}\right\}$ are the BLIMFs to be solved, $\{\omega_{k}\left(t\right)\}=\left\{\omega_{1},\ldots ,\omega_{K}\right\}$ are the center frequencies of the BLIMFs to be solved, and K is the decomposition number that is known as an a priori parameter. In Equation (1), the $\left(\delta \left(t\right)+\frac{j}{\pi t}\right)*u_{k}\left(t\right)$ part represents the transformation of the real-valued BLIMF $u_{k}\left(t\right)$ into its analytical signal. Then the analytical signals are shifted to the zero-frequency baseband, and finally, the effective bandwidth is obtained using the L2 norm of the time derivative. The quadratic penalty and Lagrangian multiplier are introduced to address Equation (1) by the augmented Lagrange L as follows:(2)$$L\left(\left\{u_{k}\right\},\left\{\omega_{k}\right\},\lambda \right)=\alpha \sum_{k=1}^{K}\|\partial_{t}\left[\left(\delta \left(t\right)+\frac{j}{\pi t}\right)*u_{k}\left(t\right)\right]e^{-j\omega_{k}t}{\|}_{2}^{2}+\|f\left(t\right)-\sum_{k=1}^{K}u_{k}{\left(t\right)\|}_{2}^{2}+<\lambda \left(t\right),f\left(t\right)-\sum_{k=1}^{K}u_{k}\left(t\right)>,$$
where α represents the balancing parameter of the data-fidelity constraint and $\lambda \left(t\right)$ denotes the Lagrangian multipliers. Equation (2) is then solved through the alternate direction method of multipliers (ADMM) approach. Using the condition that the signal is real-valued, the BLIMFs in spectral domain can be solved by
(3)$${\hat{u}}_{k}^{n+1}\left(\omega \right)=\frac{\hat{f}\left(\omega \right)-\sum_{i\neq k}{\hat{u}}_{k}\left(\omega \right)+\left(\hat{\lambda }\left(\omega \right)/2\right)}{1+2\alpha {\left(\omega -\omega_{k}\right)}^{2}},\;\omega \geq 0,$$
where the $\omega_{k}$ is calculated at the center of gravity of the corresponding mode’s power spectrum as
(4)$$\omega_{k}^{n+1}=\frac{\int_{0}^{\infty }\omega {\left|{\hat{u}}_{k}\left(\omega \right)\right|}^{2}d\omega }{\int_{0}^{\infty }{\left|{\hat{u}}_{k}\left(\omega \right)\right|}^{2}d\omega }.$$

The BLIMFs in time domain $\left\{u_{k}\left(t\right)\right\}$ are obtained from the real part of the inverse Fourier transform of the frequency spectra.

As the standard VMD has been a mature algorithm built as a function in MATLAB since 2020, the subsequent VMD procedure is based on the MATLAB function. Without a special note, all of the parameters, such as the way of initializing the center frequencies $\omega_{k}$ and the parameter α, are set as default.

### 2.2. Complex Variational Mode Decomposition

Since the VMD is designed for real-valued data, the extension method of VMD for handling complex-valued data called CVMD was proposed. CVMD divides the complex-valued signal into positive and negative frequency components and uses the relationship between the Fourier spectrum and the analytical signal to get real signals for the standard VMD algorithm.

Let $x\left(t\right)$ be the complex-valued data and $X\left(e^{j\omega }\right)$, where −π≤ω<π is the Fourier transform of $x\left(t\right)$. The complex-valued data are separated into positive and negative components by using a band-pass filter as
(5)$$H\left(e^{j\omega }\right)=\left\{\begin{array}{c} 1,\;0\leq \omega <\pi \\ 0,-\pi \leq \omega <0 \end{array}\right..$$

The following two real signals can be generated by
(6)$$x_{+}\left(t\right)=R\{F^{-1}\left[H\left(e^{j\omega }\right)X\left(e^{j\omega }\right)\right]\},$$
(7)$$x_{-}\left(t\right)=R\{F^{-1}\left[H\left(e^{j\omega }\right)X^{*}\left(e^{-j\omega }\right)\right]\},$$
where * describes the complex conjugate operation, symbol $R\left\{\cdot \right\}$ defines the real part of the complex-valued data, and $F^{-1}\left[\cdot \right]$ denotes the inverse Fourier transform. According to the properties of the analytic signals, $F^{-1}\left[H\left(e^{j\omega }\right)X\left(e^{j\omega }\right)\right]$ and $F^{-1}\left[H\left(e^{j\omega }\right)X^{*}\left(e^{-j\omega }\right)\right]$ are analytic signals, and their real parts contain all of the information. The reconstruction of the original complex-valued signal is determined by $x_{+}\left(t\right)$ and $x_{-}\left(t\right)$ as well as their Hilbert transform pairs, expressed as
(8)$$x\left(t\right)=\left(x_{+}\left(t\right)+j\hbar \left[x_{+}\left(t\right)\right]\right)+{\left(x_{-}\left(t\right)+j\hbar \left[x_{-}\left(t\right)\right]\right)}^{*},$$
where $\hbar \left[\cdot \right]$ denotes the Hilbert transform operator.

Since $x_{+}\left(n\right)$ and $x_{-}\left(n\right)$ are real-valued signals, the corresponding BLIMFs can be obtained using standard VMD. The VMD is separately embraced to the two real-valued signals. Suppose the decomposition number of the positive and negative frequency planes is $K_{+}$ and $K_{-}$, respectively. The existing methods consider $K_{+}=K_{-}=K$, so the decomposition is described as
(9)$$x_{+}\left(t\right)=\sum_{i=1}^{K}x_{i}\left(t\right),$$
(10)$$x_{-}\left(t\right)=\sum_{i=-K}^{-1}x_{i}\left(t\right),$$
where ${\left\{x_{i}\left(t\right)\right\}}_{i=1}^{K}$ and ${\left\{x_{i}\left(t\right)\right\}}_{i=-K}^{-1}$ denote sets of BLIMFs corresponding to $x_{+}\left(t\right)$ and $x_{-}\left(t\right)$, respectively.

Based on the Equations (8)–(10), the CVMD for a complex-valued data $x\left(t\right)$ is eventually expressed as
(11)$$x\left(t\right)=\sum_{i=-K,i\neq 0}^{-K}z_{i}\left(t\right),$$
where $z_{i}\left(t\right)$ represents the *i*-th complex BLIMF, which can be written as
(12)$$z_{i}\left(t\right)=\left\{\begin{array}{c} x_{i}\left(t\right)+j\hbar \left[x_{i}\left(t\right)\right],i=1,\ldots ,K\\ {\left(x_{i}\left(t\right)+j\hbar \left[x_{i}\left(t\right)\right]\right)}^{*},i=-K,\ldots ,-1 \end{array}\right..$$

## 3. Problem Statement

The VMD mathematical theory demonstrates that the algorithm is unsuitable for complex-valued signal analysis. CVMD extent VMD to complex-valued data, but here we will illustrate its limitations in decomposing the signal with a low-frequency trend. Take the complex-valued signal $s_{1}\left(t\right)=0.5e^{j2\pi \left(-8t\right)}+e^{j2\pi \left(-t+0.6t^{2}\right)}+0.8e^{j2\pi \left(16t+0.2t^{2}\right)}$ for example. The blue solid line in Figure 1 shows the frequency spectrum of $s_{1}\left(t\right)$. Three band-limited components can be seen distributed in different parts of the spectrum: the first component, $x_{1}=0.5e^{j2\pi \left(-8t\right)}$, is mainly at negative frequency; the second component, named $x_{2}=e^{j2\pi \left(-t+0.6t^{2}\right)}$, is around zero frequency; and the third one, $x_{3}=0.8e^{j2\pi \left(16t+0.2t^{2}\right)}$, is mainly at positive frequency. Note that the center frequency of the low-frequency signal $x_{2}$ is actually at the positive frequency, but there is energy spreading over negative frequencies.

In CVMD, the positive and negative frequency components of the signal are displayed on the positive and negative frequency planes, respectively. Both of the decomposition numbers of two planes are required. The decomposition number of the negative frequency plane is considered the same as that of the positive frequency plane. However, only the whole data’s decomposition number can be easily known in this example. It is unclear how many components are at the positive frequency plane or negative frequency plane without prior knowledge. Even if the correct decomposition number is known and the signals are rearranged well, the results by CVMD have energy loss in the components around zero frequency. The red dotted line in Figure 1 shows the reconstruction of the CVMD results. It can be seen clearly that signal $x_{2}$ is not recovered completely.

To further illustrate this problem, the behavior of CVMD as a filter bank is illustrated by applying the algorithm to the white Gaussian noise. Both the positive frequency and negative frequency components’ decomposition number is set as 6, for example. The equivalent filter bank of CVMD is shown in Figure 2, in which the parts near zero frequency, $x_{1+}$ and $x_{1-}$, are separated artificially by the filter. This separation goes against the data-driven concept of VMD. Although the low-frequency trend can be reconstructed by combining $x_{1+}$ and $x_{1-}$ components, it will not be recovered ideally.

## 4. Modified Complex Variational Mode Decomposition

CVMD cannot reconstruct the low-frequency trend completely because it separates the data into positive and negative frequency components artificially and hurts the parts around zero frequency. A modified VMD method is proposed to overcome the defect based on upsampling and frequency shifting.

### 4.1. MCVMD Algorithm

For real-valued data, the positive- and negative frequency components are conjugate symmetric. For complex-valued data, its positive- and negative frequency components contain different information, which is not symmetric. The idea of the proposed MCVMD method is to convert complex-valued data into real-valued data without losing information so that standard VMD can be used. This idea is similar to CVMD and CEMD, but the operation is different. The well-known property of the analytic signal utilized in the proposed method is that the analytic signal of a real-valued signal only has positive frequency spectra but contains all of the information of the origin real-valued data. Alternatively, the real part of the analytic signal has symmetrical positive and negative frequencies and contains all of the origin complex-valued analytic signal information. According to the property, if the whole spectrum of the complex-valued data are shifted to the positive frequency, then the complex-valued data become an analytic signal. The real-valued data can be obtained by taking the real part of the analytic signal and contain all of the information of the original complex-valued data. Since the signal in practical applications is almost discrete data, the proposed method is designed for discrete signals. The steps of the algorithm are listed as follows:

Consider $x\left(n\right)$, where 0≤n≤N−1 and n is an integer, as the discrete complex-valued data, and the sampling frequency rate is normalized as 1. Suppose the sampling rate satisfies the Nyquist sampling frequency. That is to say, the max signal frequency of the data do not exceed 1/2, and the minimum signal frequency is not less than −1/2.

First, the input data should be upsampled by the interpolation algorithm, among which the discrete sinc interpolation algorithm does not distort the signal as defined and is entirely reversible. However, it is often not used directly due to the boundary effects of sinc interpolation [31]. One of the simplest and most efficient methods for minimizing boundary effects is the mirror extension of the signal by half its length on each side. Accordingly, the mirror extension of the original input data are expressed as
(13)$$x_{e}=\left[x_{e1},x_{e2},x_{e3}\right],$$
where $x_{e1}$ denotes the refection sequence of the first half of the original data as
(14)$$x_{e1}\left(n\right)=x\left(\frac{N}{2}-n-1\right),\;0\leq n\leq \frac{N}{2}-1,$$
where $x_{e2}$ denotes the original sequence as
(15)$$x_{e2}\left(n\right)=x\left(n\right),\;0\leq n\leq N-1,$$
where $x_{e3}$ denotes the refection sequence of the last half of the original data as
(16)$$x_{e3}\left(n\right)=x\left(N-n-1\right),\;0\leq n\leq \frac{N}{2}-1.$$

The extended signal can be rewritten as $x_{e}\left(n\right)$, where 0≤n≤2N−1.

The discrete Fourier transform (DFT) is applied to the extended signal as
(17)$$X_{e}\left(k\right)=\sum_{n=0}^{2N}x_{e}\left(n\right)e^{-j\frac{2\pi }{2N}kn}\;,\;0\leq k\leq 2N-1,$$

This represents the spectral samples. Then, 2N samples valued at zero are inserted after the first N spectral samples. A one-dimensional vector with length 4N can be generated as
(18)$$X_{ez}=\left[X_{e1},Z,X_{e2}\right],$$
where $X_{e1}$ denotes the sequence $X_{e}\left(k\right)\;$(0≤k≤N−1), Z denotes 2N samples valued at zero, and $X_{e2}$ denotes the sequence $X_{e}\left(k\right)\;$(N≤k≤2N−1), which can be rewritten as $X_{ez}\left(k\right),\;0\leq k\leq 4N-1.$ The corresponding time–domain data can be obtained by
(19)$$x_{ez}\left(n\right)=2F^{-1}\left[X_{ez}\left(k\right)\right],0\leq k\leq 4N-1,0\leq n\leq 4N-1,$$
where $F^{-1}\left[\cdot \right]$ denotes the inverse discrete Fourier transform.

The actual upsampled result is given by removing the mirror part as
(20)$$x_{z}\left(n\right)=x_{ez}\left(n+N\right),0\leq n\leq 2N-1.$$

It is known that zero padding in the frequency domain leads to an increased sampling rate in the time domain [32]. Since the data length is doubled by zero padding, the sampling rate of $x_{z}\left(n\right)$ now becomes 2. In addition, the Fourier spectrum of $x_{z}\left(n\right)$ now has definitions from the frequency of -1 to 1 and has values from the frequency of -1/2 to 1/2.

$x_{z}\left(n\right)$ is then shifted by multiplying exponential factors $e^{j2\pi \cdot \frac{n}{4}}$ as follows
(21)$$x_{zs}\left(n\right)=x_{z}\left(n\right)e^{j2\pi \cdot \frac{n}{4}},0\leq n\leq 2N-1.$$

Since the frequency of $x_{z}\left(n\right)$ is within −1/2 to 1/2, all of the components of $x_{zs}\left(n\right)$ are within 0 to 1 after frequency shifting, which means it has only positive frequency components. Thus, $x_{zs}\left(n\right)$ becomes an analytic signal. According to the property of the analytic signals, real parts of the analytic signal contain all of the information. A real-valued signal can be deduced as
(22)$$x_{zsr}\left(n\right)=R\left\{x_{zs}\left(n\right)\right\}.$$

Since $x_{zsr}\left(n\right)$ is real valued, the BLIMFs can be obtained using standard VMD. Consider that the decomposition number is K. The mode decomposition procedure can be expressed as
(23)$$x_{zsr}\left(n\right)=\sum_{i=1}^{K}u_{i}\left(n\right)+r\left(n\right),$$
where $u_{i}\left(n\right)$ denotes the *i*-th BLIMF corresponding to $x_{zsr}\left(n\right)$, and $r\left(n\right)$ describes the residual.

After VMD, the Hilbert transform is applied to generate the corresponding complex and analytic *i*-th BLIMF as
(24)$$y_{i}\left(n\right)=u_{i}\left(n\right)+j\hbar \left[u_{i}\left(n\right)\right].$$

Note the data are upsampled and modulated, so they are not the final result of MCVMD.

The final complex BLIMFs can be obtained by demodulation and downsampling. The *i-*th upsampled and modulated complex BLIMFs $y_{i}\left(n\right)$ are shifted back by multiplying exponential factors $e^{-j2\pi \cdot \frac{n}{4}}$ as
(25)$$z_{i}\left(n\right)=y_{i}\left(n\right)e^{-j2\pi \cdot \frac{n}{4}},$$
where $z_{i}\left(n\right)$ denotes the *i-*th upsampled complex BLIMFs, which has the double sample rate of the original data. The final *i-*th complex BLIMF $q_{i}\left(n\right)$ corresponding to the origin complex data $x\left(n\right)$ can be obtained simply by downsampling.
(26)$$q_{i}\left(n\right)=z_{i}\left(2n+1\right),0\leq n\leq N-1.$$

### 4.2. MCVMD Hilbert Spectrum

The mode decomposition method is often used to get the Hilbert–Huang spectrum, which gives the time–frequency representation [33]. Specifically, mode decomposition is the first step in conventional Hilbert–Huang spectrum analysis. The Hilbert transform is then applied to the real-valued decomposed modes to get the analytic signal. Analytic signals are required to analyze each mode’s instantaneous phase and amplitude [34]. Here, the complex BLIMF from MCVMD is already analytical. Thus, the instantaneous phase and instantaneous amplitude can respectively be obtained directly by the following equations
(27)$$\emptyset_{i}\left(n\right)=arctan\left(\frac{I\left\{q_{i}\left(n\right)\right\}}{R\left\{q_{i}\left(n\right)\right\}}\right),1\leq i\leq K,\;0\leq n\leq N-1,$$
(28)$$A_{i}\left(n\right)=\sqrt{{\left(I\left\{q_{i}\left(n\right)\right\}\right)}^{2}+{\left(R\left\{q_{i}\left(n\right)\right\}\right)}^{2}},1\leq i\leq K,0\leq n\leq N-1,$$
where symbol $I\left\{\cdot \right\}$ defines the imaginary part of a complex function, $\emptyset_{i}\left(n\right)$ represents the instantaneous phase, and $A_{i}\left(n\right)$ corresponds to the instantaneous amplitude of the *i-*th complex BLIMF $q_{i}\left(n\right)$. The instantaneous frequency can be derived as
(29)$$f_{i}\left(n\right)=\frac{1}{2\pi }\frac{d\emptyset_{i}\left(n\right)}{dn}.$$

Accordingly, the Hilbert spectrum of each complex BLIMF can be obtained as
(30)$$S_{i}\left(f,n\right)=A_{i}\left(n\right).$$

The MCVMD Hilbert spectrum intuitively shows the instantaneous frequency and amplitude of each mode in the signal, which is not constrained by the uncertainty limitations. Unlike the VMD Hilbert spectrum that only contains positive frequency, the MCVMD Hilbert spectrum can represent the positive and negative frequencies. Thus, the time-varying characteristics of low-frequency trends varying from positive frequency to negative frequency or from negative to positive can also be seen.

### 4.3. MCVMD Equivalent Filter Bank

CVMD has been applied to white Gaussian noise to analyze its equivalent filter bank. In this subsection, the equivalent filter bank of MCVMD is also investigated. MCVMD is applied to Gaussian complex-valued data of 1024 randomly generated samples, and the decomposition number is set as 12. Thus, 12 complex BLIMFs and their spectra can be obtained. This process is repeated 2000 times independently, and then the resulting spectra are averaged. Figure 3 shows these averaged spectra for the 12 complex BLIMFs. It can be seen CVMD equivalent filter bank structures behave almost constant bandwidths. Compared to the equivalent filter bank of CVMD shown in Figure 2, the low-frequency components are no longer separated. Each resulting structure by MCVMD is natural and complete over the whole frequency range without artificial separation. The result exhibits the desired behavior of a filter bank, which confirms the proposed MCVMD being a natural extension of the standard VMD and outperforming the CVMD.

## 5. Results and Discussion

### 5.1. Synthetic Signal Analysis

#### 5.1.1. Example 1

In this example, the performance of MCVMD in decomposing the signal with a low-frequency component is verified. Consider the signal $s_{1}\left(t\right)=0.5e^{j2\pi \left(-8t\right)}+e^{j2\pi \left(-t+0.6t^{2}\right)}+0.8e^{j2\pi \left(16t+0.2t^{2}\right)}$ mentioned in Section 3. Suppose the sampling rate is 50 Hz and the number of samples is 150. The discrete signal can be written as $s_{1}\left(n\right)=0.5e^{j2\pi \left(-8\cdot 0.2n\right)}+e^{j2\pi \left(-0.2n+0.6\cdot {\left(0.2n\right)}^{2}\right)}+0.8e^{j2\pi \left(16\cdot 0.2n+0.2\cdot {\left(0.2n\right)}^{2}\right)}$, where n=1,2,…,150. The mode decomposition methods aim to decompose the signal into an ensemble of sub-signals to analyze the nonstationary data. Applying the MCVMD to $s\left(n\right)$, three complex BLIMFs can be obtained. The solid blue line in Figure 4 shows the spectrum of the original signal, and the red dotted line represents the spectrum of the reconstruction signal obtained by summing up these three sub-signals. Compared with the corresponding CVMD result in Figure 1, MCVMD better reconstructs the original data, especially the second components around zero frequency.

The time–domain waveforms of the three BLIMFs obtained by CVMD and MCVMD are compared in Figure 5. Let $x_{1},\;\;x_{2},\;\;and\;x_{3}$ describe the three constituent modes $-0.5e^{j2\pi \left(-8t\right)}$, $e^{j2\pi \left(-t+0.6t^{2}\right)}$, and $0.8e^{j2\pi \left(16t+0.2t^{2}\right)}$, respectively. Figure 5 shows the original modes (red solid line) and the decomposition results by CVMD (green dash-dot line) and MCVMD (blue dotted line). The real part and imaginary part of each component are displayed separately. It can be seen both CVMD and MCVMD give a proper decomposition of $x_{1}$ and $x_{3}$. However, the performance of the two methods differs in reconstructing the second mode $x_{2}$. MCVMD gives a better decomposition and reconstruction, especially in the imaginary part of $x_{2}$. This phenomenon is consistent with the results of the spectrum display.

The root mean square error (RMSE) of each decomposed mode is shown in Table 1, in which the real and imaginary parts are considered separately. From Figure 5, both CVMD and MCVMD yield improper values at the beginning and end of the signal. Since this problem is because of the original VMD algorithm and is not the point here, the RMSE results in this paper are given by removing the first and last five samples. Compared with CVMD, the RMSE of the decomposed modes by MCVMD is generally smaller. As can be seen, the RMSE of the imaginary part of the low-frequency mode by MCVMD is almost one-tenth of that by CVMD.

#### 5.1.2. Example 2

To further prove the proposed method’s ability to analyze the signal’s time–frequency characteristics, we give another more complex example of MCVMD developing the bi-directional Hilbert spectrum. The synthetic signal has two modes as follows
(31)$$s_{2}\left(t\right)=cos\left(0.2t\right)\cdot e^{j2\pi \left(-t+0.5t^{2}\right)}+0.2t\cdot e^{j2\pi \left(28t-sin\left(\pi t\right)\right)},$$
where both amplitudes and frequencies are time varying. The signal length is 5 s, and the sampling rate is 100 Hz. The amplitude of the first mode varies as the cosine function, and the frequency varies linearly between -1 Hz and 4 Hz. The amplitude of the second mode increases linearly with time, and the frequency varies as the sine function. Hilbert transform is used to get the two modes’ instantaneous amplitudes and frequencies. Figure 6a shows the actual Hilbert spectrum of the two simulation sub-signals, which is used for comparison later.

The proposed MCVMD method is applied to $s_{2}\left(t\right)$. After that, we use Equations (27)–(30) to get the corresponding Hilbert spectrum shown in Figure 6b. The MCVMD Hilbert spectrum presents the time–frequency characteristics and shows the two modes’ instantaneous amplitude and frequency as the actual Hilbert spectrum. The corresponding results by CVMD are shown in Figure 6c. As can be seen, the instantaneous amplitude of the low-frequency mode by CVMD behaves differently from the actual mode.

The RMSE of each mode’s instantaneous frequency and amplitude are presented in Table 2. It shows that both the instantaneous frequency and amplitude’s RMSE by MCVMD are smaller than by CVMD, especially in the low-frequency mode. It can be concluded that MCVMD not only gives a better decomposition of the multi-component signal, but also gives a better estimation for instantaneous frequency and instantaneous amplitude with the corresponding Hilbert spectrum.

#### 5.1.3. Example 3

In this subsection, we discuss the prior requirements of the proposed method on the number of decompositions. The results obtained in Example 2 rely on the precise knowledge of the decomposition number K. In the MCVMD method, the correct decomposition number is K=2 for the entire frequency range. In the CVMD method, the appropriate decomposition number is $K_{+}=3$ for the positive frequency range and $K_{-}=1$ for the negative frequency range. Numerical experiments are presented to find out the influence of incorrect decomposition numbers.

Considering the same simulation scenario as Example 2, we set the decomposition number K from 1 to 6 in MCVMD. Figure 7a shows the second mode is lost for the case K=1, and when K>1, all of the modes can be obtained as in Figure 7b,c. Although one component may be divided into more than one mode when K>2 as shown in Figure 7c, the separated modes can be summed up to recover the component without loss. This separation is not like the separation by the filter in CVMD, which is natural and data driven. The RMSEs of each decomposed mode versus decomposition number are summarized in Figure 8a, in which the RMSEs of the real and imaginary parts of each mode are averaged. The performance remains superior as long as the decomposition number exceeds two. Thus, the requirement of the decomposition number for MCVMD, in this case, is no less than two. This prior requirement is not difficult to meet in practical applications.

For CVMD, there are many more corresponding decomposition schemes to be considered for the same situation, as shown in Table 3. That is because the CVMD needs to consider the positive and negative frequencies separately. In Reference [26], $K_{-}$ and $K_{+}$ are suggested to be the same number. Here, if we set $K_{-}$ and $K_{+}$ both equal to 2, the CVMD results are obtained in Figure 9a, which can be seen the second mode is lost. The two modes can be obtained completely if we set $K_{-}=1$ and $K_{+}=3$, as shown in Figure 9b. Numerical experiments of the 15 schemes show that modes can be obtained completely only when $K_{-}\geq 1$ and $K_{+}\geq 3$. Precisely, only 3 of the 15 schemes succeed to recover the two modes, with the RMSE less than 0.11, as shown in Figure 8b. Thus, the prior requirement of CVMD is much more challenging to meet.

### 5.2. Real-World Data Analysis

#### 5.2.1. Analysis of Float Drift Data

We apply the proposed method to a real-world complex-valued data set to demonstrate its ability further. The data were taken during the Eastern Basin experiments, which can be downloaded from the World Ocean Circulation Experiment Subsurface Float Data Assembly Center (WFDAC) http://wfdac.whoi.edu, accessed on 23 January 2022 [24]. It contains the position record of a subsurface oceanographic float deployed in the North Atlantic Ocean to track the trajectory of salty water flowing from the Mediterranean Sea [35]. The looping trajectories of the chosen float are shown in Figure 10a (blue solid line). Here, the east and north displacements can be regarded as the real and imaginary part of the complex-valued data. The application of CVMD for analyzing the float drift data are illustrated in Reference [26]. Each BLIMF associated with the real and imaginary parts indicates a different physical path during vortex evolution. However, to characterize low-frequency trends, CVMD must combine the positive frequency BLIMFs and negative frequency BLIMFs. It can be seen from Reference [26], even if the signal is rearranged well, the low-frequency physical pathway given by CVMD is quite different from the actual situation, which fails to describe the potential physical characteristic.

The complex-valued input data with 548 samples are shown in Figure 11 (top row), where the real part (blue line) and the imaginary part (red line) are displayed, respectively. The proposed MCVMD is applied to the data to analyze their principal modulated oscillations. The composition number is set as 4. Thus, four modes are shown as time plots (Figure 11) and 2D graphical plots (Figure 10). The 2D plot represents the drifting float’s modulated oscillations or rotating modes. The displacement east is the real part of the resulting BLIMF, and the displacement north is the imaginary part of the resulting BLIMF. The red solid line in Figure 10a shows the low-frequency drift trajectory of the buoy, and the other three decomposed modes in Figure 10b–d show its specific rotation characteristics. We also apply CVMD to the same dataset. The yellow dotted line in Figure 10 shows the decomposed rotating modes by CVMD. It can be seen that the low-frequency drift trajectory of the buoy in Figure 10a,e by CVMD does not fully show the true characteristics. Again, that is because CVMD uses band filters and loses the low-frequency information. This example demonstrates the ability of MCVMD to analyze physically meaningful real-world data while also verifying the successful extension of VMD to complex-valued data.

#### 5.2.2. Analysis of Wind Data

In this subsection, we apply the proposed method to the real-world wind data, whose measurements consist of wind direction $\varphi \left(t\right)$ and wind speed $v\left(t\right)$, which can be represented as a complex variable: $x\left(t\right)=v\left(t\right)e^{j2\pi \varphi \left(t\right)/360}\;$ [30]. If we apply VMD to speed and direction data, respectively, the signal components’ positive and negative frequencies cannot be distinguished [25,26]. Thus, the complex extension of VMD is required to analyze the complex-valued form of wind data. We apply the CVMD and MCVMD methods to the wind data recorded by the Automated Weather Observing System (AWOS), which can be downloaded from http://mesonet.agron.iastate.edu/ASOS/, accessed on 23 January 2022. Figure 12 shows the time series of the wind speed and wind direction. The decomposition number K is set as 8 in MCVMD, and the positive decomposition number $K_{+}$ and negative decomposition number $K_{-}$ in CVMD are both set as 4. Figure 13 shows the complex-valued wind data decomposition results by MCVMD and CVMD, where the real part (blue line) and the imaginary part (red line) are displayed, respectively. From top to bottom corresponds to the mode with negative to positive frequency. In CVMD, the positive and negative frequency modes are obtained separately, in which the first four are the negative frequency, and the last four are positive. Summing up all of the decomposed modes, we can reconstruct the CVMD and MCVMD results. Figure 14 compares the spectrum of the original data, the reconstruction of CVMD results, and the reconstruction of MCVMD results. It can be seen that there is energy loss in the low-frequency component by CVMD, while MCVMD gives a better reconstruction. Similarly, that is because the CVMD uses band filters to separate the original data and go against the ‘data-driven’ idea of VMD. Thus, the low-frequency decomposed mode by CVMD cannot represent the wind’s real low-frequency trend ideally. MCVMD gives better decomposition and reconstruction of the signals with low-frequency components.

## 6. Conclusions

This paper presents a modified complex variational mode decomposition (MCVMD) method for decomposing a complex-valued signal into an ensemble of complex band-limited intrinsic mode functions (BLIMFs). The proposed approach extends standard variational mode decomposition (VMD) to complex-valued data. In contrast to the existing complex variational mode decomposition (CVMD) method, we refrain from separating the data into positive and negative frequency planes, applying VMD twice, and rearranging the positive and negative frequency parts. Instead, we upsample and modulate the input data to have only positive frequency components to apply the standard VMD. The properties of analytical signals are used to get the final complex-valued BLIMFs. Compared with CVMD, MCVMD has the following advantages. (1) The low-frequency components can be decomposed adaptively like other frequency parts, without artificial separation. (2) The MCVMD gives better decomposition and reconstruction of complex-valued signals with lower RMSE, especially of the low-frequency component. (3) VMD is only used once, resulting in a minor computation and less prior requirement for the decomposition number. (4) It is unnecessary to rearrange the decomposition results to reconstruct the low-frequency mode. The numerical experiment illustrates that the proposed MCVMD method behaves as an ideal filter bank structure over the whole positive and negative frequency plane. Applications in synthetic and real-world complex-valued signals demonstrate the effectiveness of the proposed MCVMD method.

## Figures and Tables

**Figure 1 sensors-22-01801-f001:**
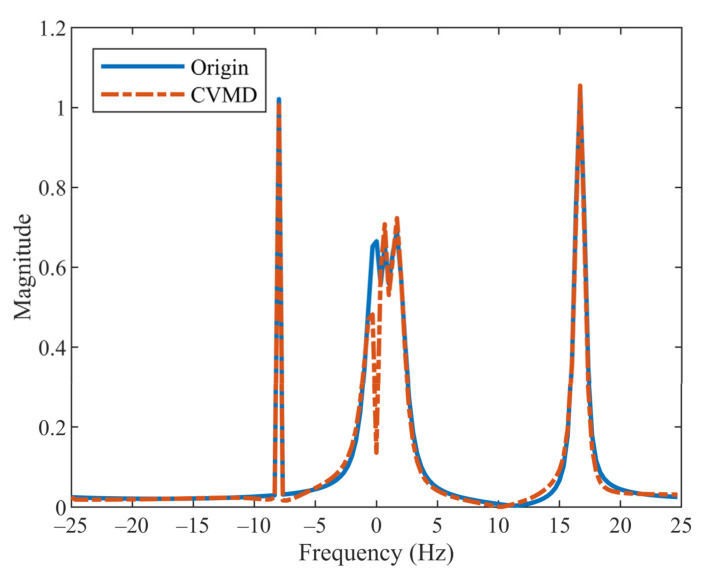
Spectrum of the original signal (blue solid line) and the reconstruction signal by CVMD (red dash-dot line).

**Figure 2 sensors-22-01801-f002:**
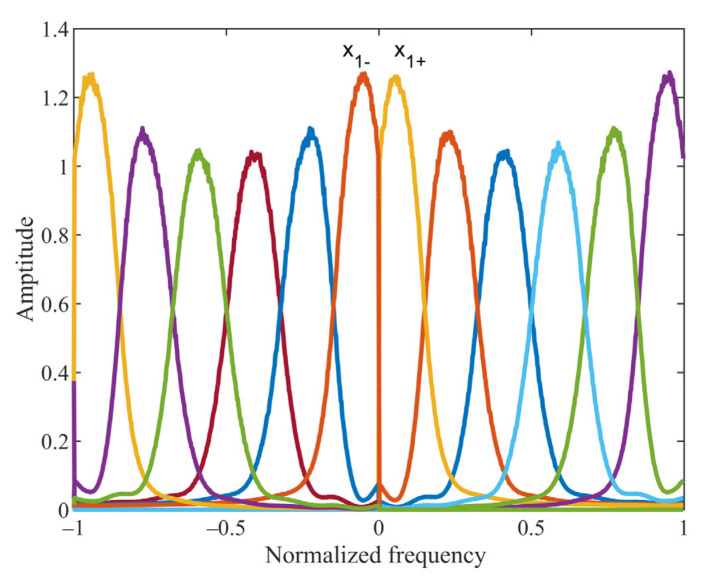
CVMD equivalent filter banks.

**Figure 3 sensors-22-01801-f003:**
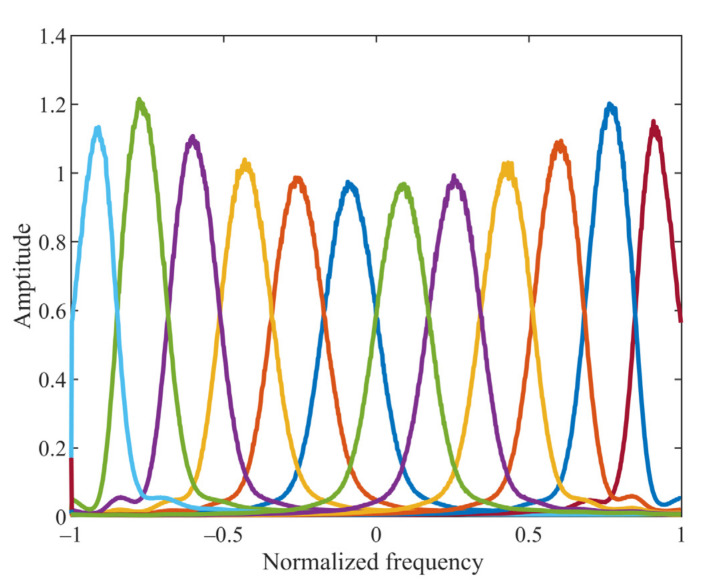
MCVMD equivalent filter banks.

**Figure 4 sensors-22-01801-f004:**
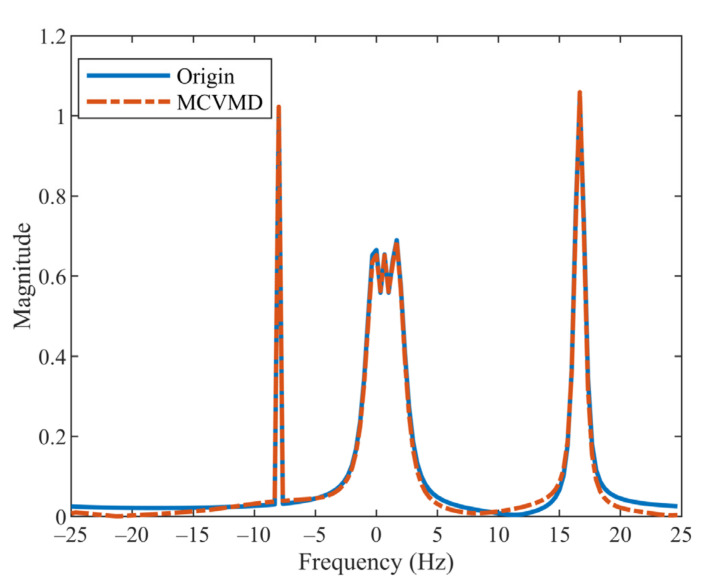
Spectrum of the original signal (blue solid line) and reconstruction signal by MCVMD (red dash-dot line).

**Figure 5 sensors-22-01801-f005:**
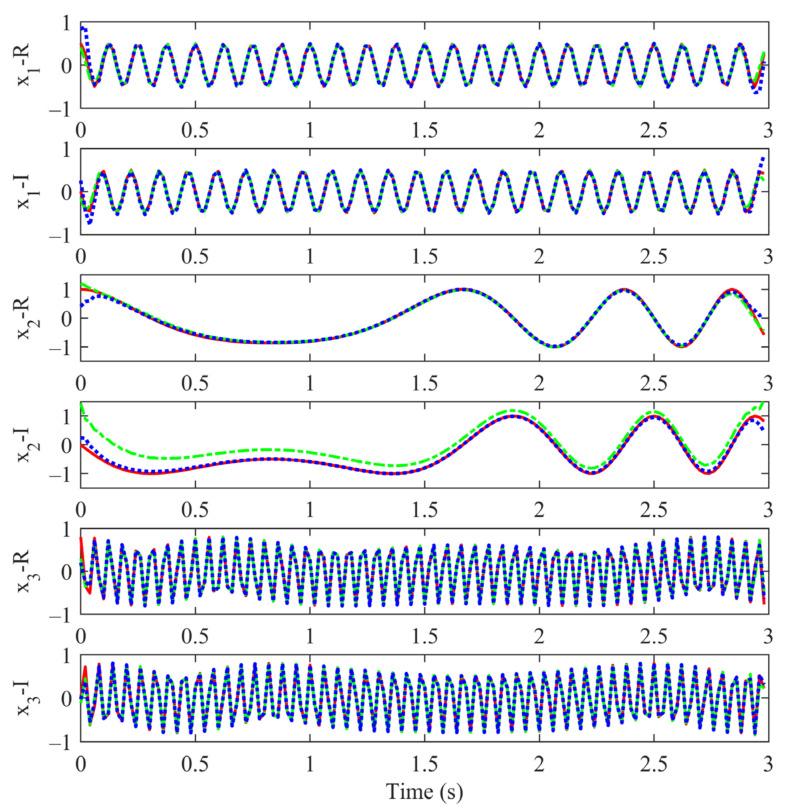
The original signal (red solid line), CVMD decomposed modes (green dash-dot line), and MCVMD decomposed modes (blue dotted line). From top to bottom: the real part of $x_{1}$, the imaginary part of $x_{1}$, the real part of $x_{2}$, the imaginary part of $x_{2}$, the real part of $x_{3}$, and the imaginary part of $x_{3}$.

**Figure 6 sensors-22-01801-f006:**
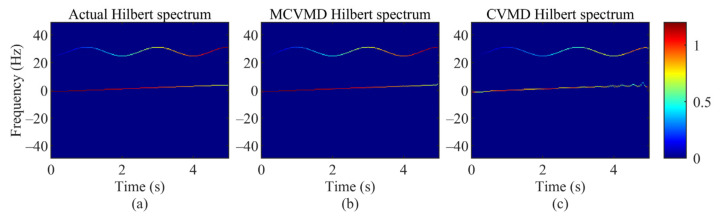
The Hilbert spectrum: (**a**) the actual Hilbert spectrum, (**b**) the MCVMD Hilbert spectrum, and (**c**) the CVMD Hilbert spectrum.

**Figure 7 sensors-22-01801-f007:**
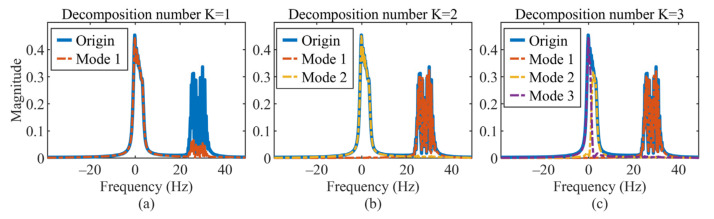
Spectrum of the original signal (blue solid line) and the decomposed modes (dotted line): (**a**) decomposition number K=1, (**b**) decomposition number K=2, and (**c**) decomposition number K=3.

**Figure 8 sensors-22-01801-f008:**
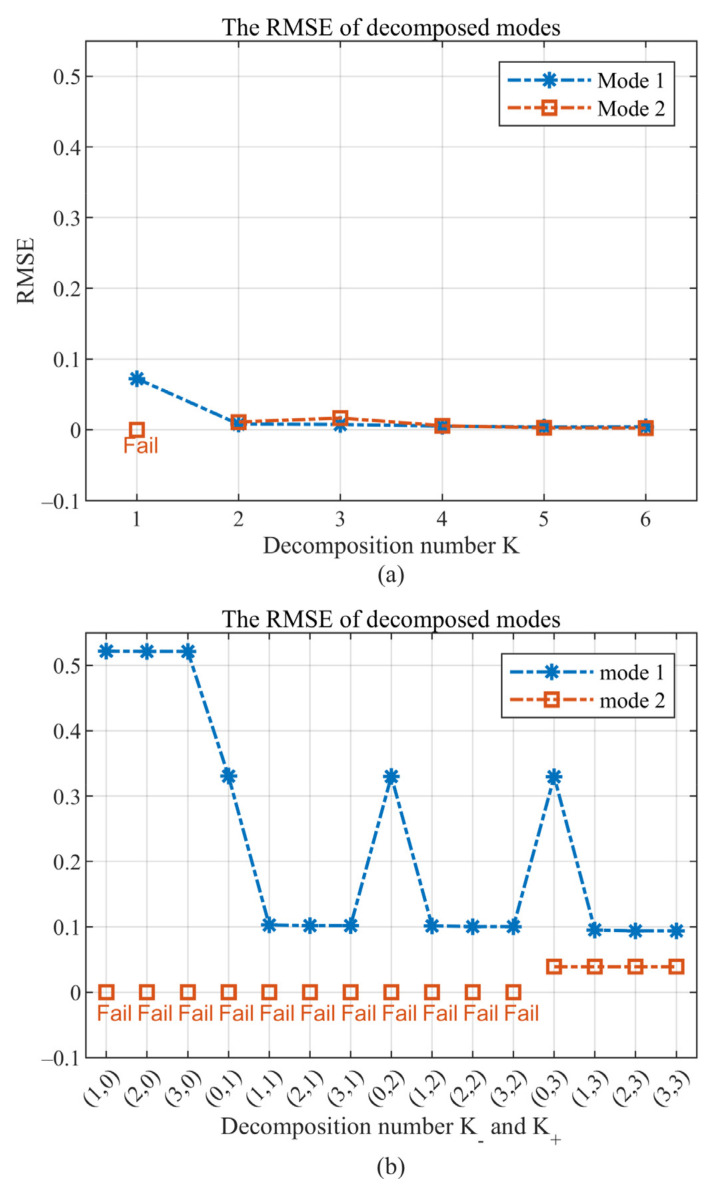
The RMSE of mode 1 (blue line) and mode 2 (red line) versus the decomposition number: (**a**) MCVMD results and (**b**) CVMD results.

**Figure 9 sensors-22-01801-f009:**
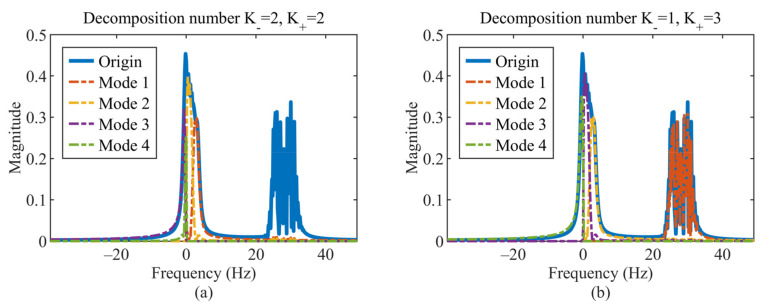
Spectrum of the original signal (blue solid line) and the decomposition modes (dotted line): (**a**) decomposition number $K_{-}=2$ and $K_{+}=2$, and (**b**) decomposition number $K_{-}=1$ and $K_{+}=3$.

**Figure 10 sensors-22-01801-f010:**
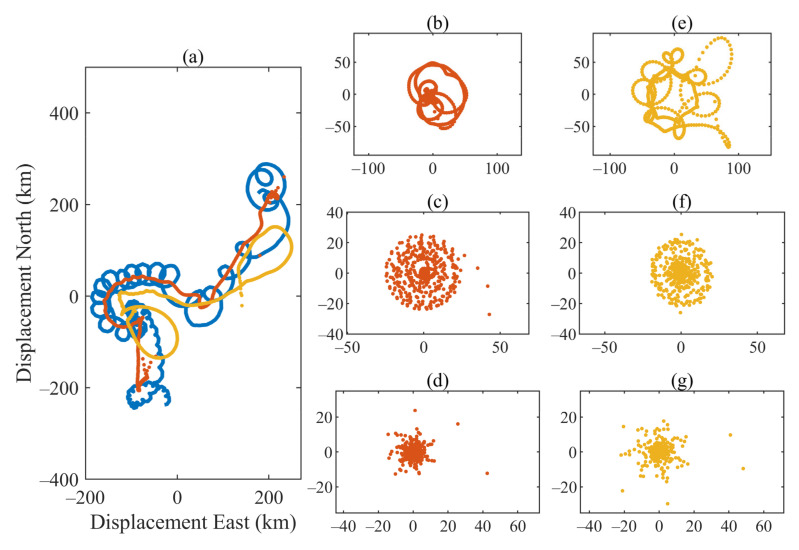
The trajectories of the original float data (blue solid line), its MCVMD results (red dotted line), and CVMD results (yellow dotted line): (**a**) original trace, MCVMD decomposed mode 1 and CVMD decomposed mode 1; (**b**) MCVMD decomposed mode 2,;(**c**) MCVMD decomposed mode 3; (**d**) MCVMD decomposed mode 4; (**e**) CVMD decomposed mode 2; (**f**) CVMD decomposed mode 3; and (**g**) CVMD decomposed mode 4.

**Figure 11 sensors-22-01801-f011:**
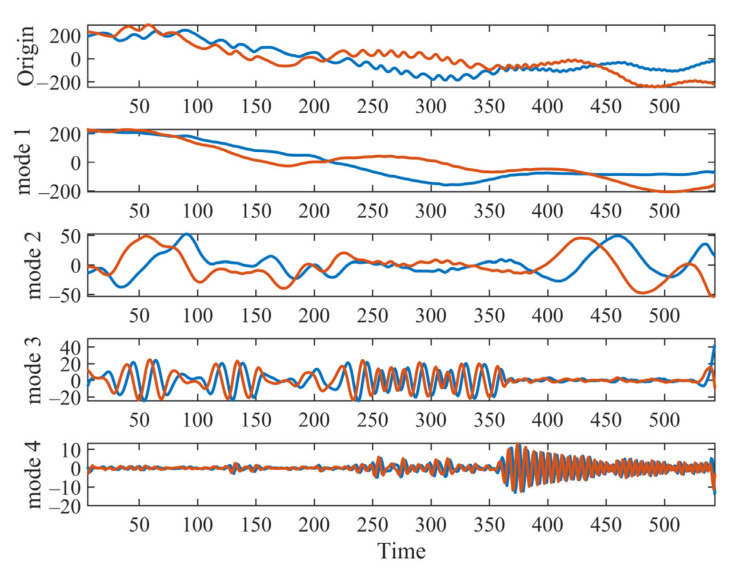
The time–domain waveform of the original drifting float signal and its MCVMD decomposition. From top to bottom: original signal, decomposed mode 1, decomposed mode 2, decomposed mode 3, and decomposed mode 4.

**Figure 12 sensors-22-01801-f012:**
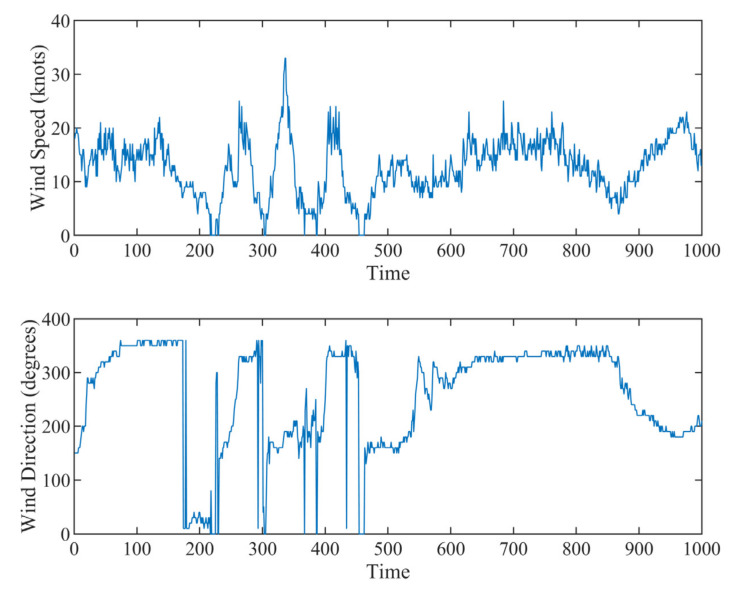
Time series of wind speed and direction.

**Figure 13 sensors-22-01801-f013:**
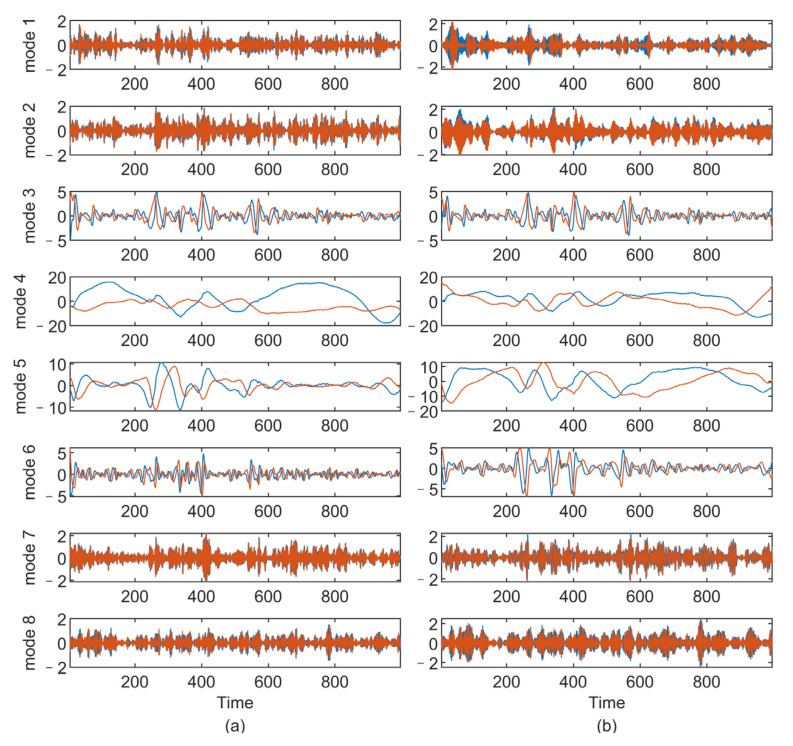
The decomposition results of the complex-valued wind data: (**a**) MCVMD results and (**b**) CVMD results.

**Figure 14 sensors-22-01801-f014:**
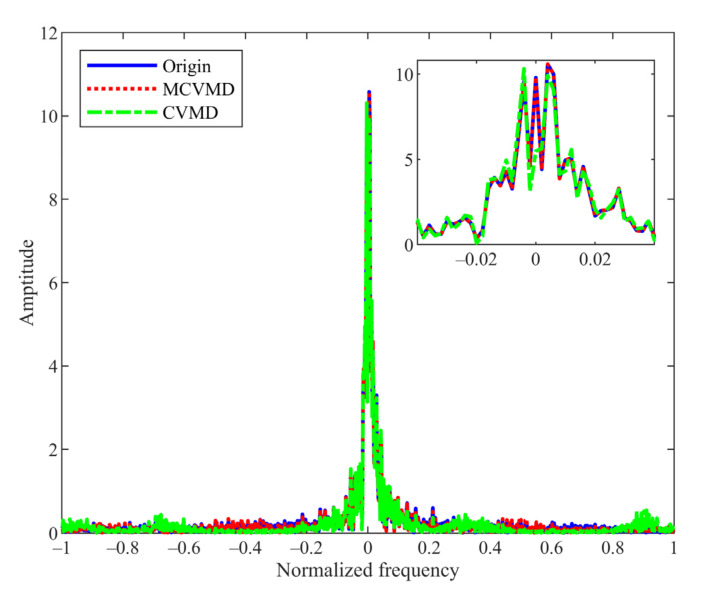
Spectrum of the original wind data (blue solid line), the reconstruction signal of MCVMD results (red dotted line), and the reconstruction signal of CVMD results (green dash-dot line).

**Table 1 sensors-22-01801-t001:** The RMSE of the decomposed modes by CVMD and MCVMD.

	$x_{1}-R^{\;1}\;$	$x_{1}-I\;$	$x_{2}-R\;$	$x_{2}-I\;$	$x_{3}-R\;$	$x_{3}-I\;$
CVMD	0.020	0.020	0.042	0.334	0.012	0.015
MCVMD	0.010	0.013	0.027	0.033	0.005	0.006

^1^ $x_{1}-R$ denotes the real part of mode $x_{1}$, and $x_{1}-I$ denotes the imaginary part.

**Table 2 sensors-22-01801-t002:** The RMSE of the instantaneous frequency and amplitude by CVMD and MCVMD.

	Mode 1-F ^1^	Mode 1-A	Mode 2-F	Mode 2-A
**CVMD**	0.0538	3.3954	0.1202	0.3566
**MCVMD**	0.0149	0.9334	0.0104	0.0780

^1^ Mode 1-F and Mode 1-A denote the instantaneous frequency and amplitude of mode 1, respectively.

**Table 3 sensors-22-01801-t003:** Decomposition schemes for CVMD.

	$K_{-}=0\;$	$K_{-}=1\;$	$K_{-}=2\;$	$K_{-}=3\;$
$K_{+}=0$	-	(1,0)	(2,0)	(3,0)
$K_{+}=1$	(0,1)	(1,1)	(2,1)	(3,1)
$K_{+}=2$	(0,2)	(1,2)	(2,2)	(3,2)
$K_{+}=3$	(0,3)	(1,3)	(2,3)	(3,3)

## Data Availability

The data that support the findings of this study are available from the corresponding author upon reasonable request.

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
