# Peer review of "A Modified Complex Variational Mode Decomposition Method for Analyzing Nonstationary Signals with the Low-Frequency Trend"

_sensors, 2022, doi:10.3390/s22051801_

Round 1

Reviewer 1 Report

There are many “modified - CVMD” versions that are being published due to the recent increase in the interest of VMD in the community. Though the benchmarks of three examples and a real-world dataset demonstrate better MCVMD decomposition at low frequency, the manuscript doesn’t satisfactorily capture how these modifications are significant or the significance of the problem of interest and the proposed solution. More examples or discussions in the introduction can be useful in this regard.

Benchmarking with other CVMD implementations (few existing tools in the literature) on a few more real-world datasets with this implementation can give clarity on the significance of the MCVMD and is also useful for readers to adopt this implementation.

Reviewer 2 Report

The authors proposed MCVMD for analyzing nonstationary signals with low-frequency content. The manuscript is well-articulated and easy to follow. However, the authors could not clarify the following:

  1. I was not satisfied with the introduction on why would authors use MCVMD when there are other mathematically robust methods are available?
  2. L36-47 is not correct. I do not understand what the authors are trying to say. It is strongly suggested to rephrase the lines with appropriate citations. 
  3. The authors did not validate the proposed method on any real-world data. Is there any specific reason for it? please clarify.
  4. Do the proposed method apt for dynamically frequency change environments?  Clarify.
  5. It is suggested to cite the following articles to further enrich the quality and readability of the paper

(a)  L. Stankovic, M. Brajovic, and D. P. Mandic, "Time-frequency decomposition of multivariate multicomponent signals,"Signal Processing, vol. 142, pp. 468-479, 2018.

(b) Sony, S. 2021. “Towards multiclass damage detection and localization
using limited vibration measurements.” Ph.D. thesis, Dept. of Civil
and Environmental Engineering, Univ. of Western Ontario, Canada.

(c) Huang, G., Su, Y., Kareem, A., and Liao, H. (2016). Time-Frequency Analysis of Nonstationary Process Based on Multivariate Empirical Mode Decomposition. Journal of Engineering Mechanics, 142(1), 04015065.

Round 2

Reviewer 1 Report

The suggestions are incorporated and corresponding modifications are done to the manuscript. I accept the manuscript in the current form.

Reviewer 2 Report

The authors have incorporated all the changes and clarified all the comments. I recommend this article for publication